# Impact of the Menstrual Cycle on Physical Performance and Subjective Ratings in Elite Academy Women Soccer Players

**DOI:** 10.3390/sports12010016

**Published:** 2024-01-03

**Authors:** Etienne Juillard, Tom Douchet, Christos Paizis, Nicolas Babault

**Affiliations:** 1Dijon Football Côte d’Or (DFCO), 17 Rue du Stade, 21000 Dijon, France; e.juillard1295@gmail.com (E.J.); tom.douchet@gmail.com (T.D.); 2Center for Performance Expertise, CAPS, U1093 INSERM, Sport Science Faculty, University of Bourgogne-Franche-Comté, 3 Allée des Stades Universitaires, BP 27877, CEDEX, 21078 Dijon, France; christos.paizis@u-bourgogne.fr

**Keywords:** performance, monitoring, microcycle, youth

## Abstract

Our study aimed to combine psychological and physical factors to explore the impact of the menstrual cycle on performance in elite academy women soccer players through weekly monitoring. Eighteen elite academy women soccer players were monitored. Players reported daily through an online anonymous survey if they were in menstruation. Players answered the Hooper Questionnaire daily, performed an Illinois Agility Test (IAT) twice a week, and rated their perceived exertion (RPE) after every training session. Tests were associated with a complete menstrual cycle reported through the online anonymous survey to determine the two weeks of the follicular phase and the two weeks of the luteal phase. Of the 18 players, 10 completed all requirements and were retained for analyses. IAT did not show significant differences throughout the menstrual cycle (*p* = 0.633). Fatigue (*p* = 0.444), Stress (*p* = 0.868), Sleep (*p* = 0.398), DOMS (delayed onset muscle soreness; *p* = 0.725), and Hooper Index (*p* = 0.403) did not show significant differences either. RPE was also comparable across the cycle (*p* = 0.846). Our results failed to demonstrate that hormonal variation during the menstrual cycle influenced psychological and physical markers of performance.

## 1. Introduction

The influence of the menstrual cycle on performance is one of the most controversial, debated topics in sports science. Some authors have demonstrated that the estrogen secreted during the follicular phase positively influenced sports performance [1,2,3,4,5,6] due to reduced cellular catabolism [1,2]. This secreted hormone could favor muscle growth by the end of the follicular phase when its concentration is the highest [3]. Moreover, comparisons between resistance training during the follicular and the luteal phases showed increased results in strength production in favor of the follicular phase [4,5,6]. Furthermore, estrogen concentration was also shown to positively influence the aerobic capacity [7,8,9]. The estradiol derivative of the estrogen hormone seems to favor lipids’ oxidative capacity, decreasing reliance on glycogen during aerobic exercises [7]. In addition, studies have demonstrated the ability of estrogen secretion to decrease oxidative damage and increase ventilatory capacity [8,9], alongside its role in the protection and stabilization of muscular cells [3,9].

However, these results have been challenged by authors who have criticized the determination methods of the different menstrual cycle phases [10]. These authors demonstrated a negative correlation between estrogen concentration and strength production when menstrual cycles were determined clinically with predictor kits analyzing the colorimetry of luteinizing hormone concentration in urine [10]. This method has been in use since 1986 to facilitate the determination of the ovulation phase [11]. Furthermore, aerobic capacities were not influenced by hormonal fluctuations [10]. Still, a recent survey showed that 97% of women reported painful symptoms during the menstrual cycle. Some 67% of women also perceived that the menstrual cycle negatively influenced their physical performance [12]. It was shown that athletes’ perception of performance was associated with overtraining syndrome and injury risk and should therefore be taken into account [13]. These results emphasize the complex relationship between the menstrual cycle and performance.

Elite women soccer players usually cover 10.3 km per match and perform multiple explosive actions requiring both great levels of aerobic and strength capacities [14,15]. The impact of the menstrual cycle on performance in such a population should, therefore, be explored. However, the scientific literature is still scarce concerning its impact on elite soccer players [16], or is inconclusive [17,18,19], and no study has attempted to combine physical and psychological factors. A previous study has analyzed the menstrual cycle’s impact on performance using its different phases (early follicular, late follicular, and mid-luteal phase) [16]. However, in soccer, training periodization is often built on a weekly basis [20,21]. This study therefore proposed to explore the impact of each week of each phase (follicular and luteal) of the menstrual cycle on performance. Considering the impact of subjective ratings on performance and weekly performance fluctuations, our study aimed to combine subjective and physical factors to explore the impact of the menstrual cycle on performance in elite academy women soccer players through weekly monitoring. Subjective ratings and performance were evaluated through the rate of perceived exertion (RPE) and Hooper Questionnaire and agility, respectively. We hypothesized that players would feel improvement their subjective feeling and physical performance in the second week of the follicular phase.

## 2. Materials and Methods

### 2.1. Participants

Eighteen elite academy women soccer players (age = 18.7 ± 1.1 years, height = 163.6 ± 6.3 cm, body mass = 59.4 ± 7.8 kg, body fat = 23.1 ± 2.5%) of the U23 team of the elite French soccer club of Dijon Football Côte d’Or (DFCO) were recruited for this study (central defender: n = 4; fullback: n = 3; central midfielder: n = 5; wide midfielder: n = 4; forward: n = 2). Only participants completing every session during the protocol were included in the study. Over the last 6 months prior to the protocol, players who reported irregular cycles (absence, or more than one-week delay) were not included in the study, nor were those who used pills as contraceptive methods due to their impact on the menstrual cycle [22,23,24]. Participants were instructed to maintain their regular nutritional status. All participants were notified of the research protocol, benefits, and risks before providing written informed consent in accordance with the declaration of Helsinki. Approval of the study was obtained from the local ethics committee. The sample size was calculated a priori using G*Power (version 3.1.9.6, free software available at https://www.psychologie.hhu.de/arbeitsgruppen/allgemeine-psychologie-und-arbeitspsychologie/gpower.html, accessed on 10 October 2021) using the following values: effect size of 0.3, power of 0.8, and probability error of 0.05. A sample size of 17 volunteers was indicated.

### 2.2. General Design

The present study was presented during the strength and conditioning society congress in Sao Paulo in 2022 [25]. This study was conducted during the 2021–2022 in-season for nine consecutive weeks before the winter break, between the 13th and 21st week of the season. The first week was used for familiarization. The eight experimental weeks were used to ensure that all players had a complete menstrual cycle (Figure 1). Throughout the eight monitored weeks, players reported daily through an online anonymous menstrual cycle questionnaire (MCQ) whether they were menstruating. Through the MCQ, participants indicated the presence/absence of bleeding. The start of the menstrual cycle was therefore accounted for on the first day when the player reported bleeding. Players were anonymized using a personalized four-digit code only known by the player herself and one experimenter not included in data collection. The follicular phase was accounted for as soon as the players were reported to be in menstruation. As demonstrated in previous studies [22], the follicular and luteal phases are separated by the ovulation phase, occurring 14 days after the onset of bleeding. After that, 14 days were also counted for the luteal phase, bringing the menstrual cycle to a total duration of 28 days. Assuming each participant had a regular cycle without variation, each phase was divided into two 7-day weeks. Consequently, the participants’ menstrual cycles were divided into two phases, each divided into two weeks for the follicular phase (Week 1 and Week 2 for FP) and two weeks for the luteal phase (Week 1 and Week 2 for LP). Only data collected throughout the complete menstrual cycle (starting after players reported bleeding and for 4 weeks) were considered for statistical analysis. Data outside of these four consecutive weeks were not analyzed.

To obtain a similar readiness level, the eight standardized training weeks throughout the protocol were strictly identical in content, duration and intensity. Therefore, training sessions from MD+2 to MD−2 were identical (Table 1). Briefly, two training sessions were performed on the second day after the match (MD+2) with a recovery-oriented session in the morning (60 min), and a strength-oriented gym-based training session in the afternoon (60 min). On MD−4, strength development was sought through a 110 min field training session. On MD−3, aerobic development was sought using a 90 min field training session. A speed-oriented session was used on MD−2 for 75 min. Players were tested daily to assess their subjective rating, and twice a week on MD+2 and MD−2 to assess their physical agility performance according to their menstrual cycle phase. Tests consisted of a Hooper questionnaire and Illinois Agility Test [26].

### 2.3. Testing Procedure

Throughout the protocol, players had to complete daily the Hooper Questionnaire. It aimed to obtain a subjective insight into individuals’ fitness level [13]. It was completed on their respective cellphone in the morning, at least one hour before the training session onset, to limit teammates’ influence. They provided their subjective feeling of sleep quality the previous night, as well as ratings of fatigue, stress, and delayed onset muscle soreness (DOMS). Each response was rated on a seven-point Likert scale, with responses ranging from “very, very good = 1” to “very, very bad = 7” for sleep, and from “very, very low = 1” to “very, very high = 7” for fatigue, stress, and DOMS. The Hooper Index (HI) was the summation of the four ratings [13]. The average of all the scores obtained during each week was calculated to obtain a single value for each week of the menstrual cycle for each variable.

On MD+2 and MD−2, just after a 15 min standardized FIFA 11+ warm-up [27], players participated in an Illinois Agility Test (IAT) to obtain an insight into their physical performance. The IAT [28] time was measured using timing gates (Witty system, Microgate, Bolzano, Italy). Players had two trials, with trials interspersed with 2 min of passive recovery. The best value was used for analyses [29]. The average of the best results of each day was calculated to obtain a value for each week of the menstrual cycle. Finally, after these experimental training sessions, players rated their perceived exertion (RPE) using a Borg CR-10 scale [30]. RPE was given on individuals’ cell phones between 15 to 30 min after the end of the session to ensure that the perceived effort value referred to the whole session rather than the most recent exercise intensity. [30]. The players had to answer the following question: “How hard was the training session?” The aim of answering on smartphones was to prevent players from giving the same value as their partners. Session-RPE (sRPE) was also calculated as the product of RPE and the training session’s duration to quantify internal and external load [31,32]. In addition, training duration was measured using a stopwatch activated at the start of the session and stopped at the end.

### 2.4. Statistical Analysis

Statistical analyses were conducted using JASP (version 0.14, JASP Team 2020, University of Amsterdam, available free at https://jasp-stats.org/download/ (accessed on 9 December 2022). Results are presented as mean values (± standard deviation, SD). Sphericity was examined by conducting Mauchly’s test. Two-way ANOVAs with repeated measurements (phase x week) were conducted with phase = follicular vs. luteal, and week = week 1 vs. week 2. Parametric tests were conducted to assess any localized effect. For each ANOVA, the effect sizes were calculated using the partial eta-squared ($\eta_{p}^{2}$) with values being considered small (<0.06), moderate (0.06–0.15), or large (>0.15) [33]. Statistical significance was set at *p* < 0.05.

## 3. Results

Out of 18 participants, 10 participants had a complete cycle analyzed (tests, questionnaire and all training sessions) and were therefore considered for analyses. The other eight participants were lost due to at least one missed training session (injury (n = 2), sickness (n = 3)), or not filling in the MCQ correctly (n = 2).

Training session variables of volume and intensity, as monitored with duration, RPE, and sRPE, did not show significant differences (Table 2). Throughout the menstrual cycle, IAT did not show any significant difference (Table 2). Furthermore, the Hooper questionnaire, as shown by Fatigue, Stress, Sleep, and DOMS, and the Hooper Index did not show significant differences either (Table 2).

## 4. Discussion

Our study aimed to combine psychological and physical factors to explore the impact of menstrual cycles on performance in elite academy women soccer players through weekly monitoring. Contrary to our hypothesis, our results did not demonstrate any increased physical performance or psychological subjective feeling during the second week of the follicular phase. In contrast, there was no significant difference throughout the menstrual cycle.

Our results did not demonstrate any difference in physical performance, as attested by the IAT. This test has been described as a valid tool to assess capacity for change of direction [34], and has already been proven valid for the detection of physical fluctuations in elite soccer players [26]. Indeed, this test requires quick changes of direction, and therefore the production of accelerations and decelerations. These qualities depend on the players’ ability to generate maximal strength levels in the shortest time possible. This directly refers to the rate of force development [35]. However, the scientific literature is equivocal on the impact of the menstrual cycle on these qualities. Some studies have reported strength fluctuations, with either increased [4,6] or decreased [4,5] levels during the follicular and luteal phases, respectively. Yet, using field tests as in our protocol, the authors demonstrated the absence of any impact of the menstrual cycle on countermovement jumps, and 20 m and 30 m sprint performances [18,36]. Furthermore, the comparison of knee extensors’ isokinetic concentric and eccentric contractile capacities in the follicular and luteal phases did not show any difference [37]. Thus, the absence of significant difference in our data is consistent with these last findings, since the contractile capacities of athletes’ lower body muscles dictate their acceleration and deceleration capabilities, making them paramount for IAT performance.

Furthermore, this study did not show any significant difference in Hooper questionnaire responses. In agreement with a previous study, sleep quality was not impacted by hormonal fluctuations [38]. However, using a different sleep monitoring scale than the Hooper questionnaire, another study that analyzed more than 395 menstrual cycles demonstrated that sleep quality could be affected during the late luteal phase (the last 5 days of the luteal phase) [39]. Nevertheless, consistent with our results, when combined with daily exercise or stress, these results were not significant [39]. Moreover, the fatigue index did not demonstrate significant differences between the follicular and luteal phases. Our results therefore contradict those in the existing literature. Fatigue is defined as the incapacity to sustain a given task to the same extent as previously performed, for example, the incapacity to lift a certain load during strength sessions [40]. According to a previous study [41], hormonal fluctuation during the follicular phase impacts neuromuscular function through increased fatigability. As mentioned above, the highest glycogen concentration during the luteal compared to the follicular phase [42] could be compensated for by the estrogen hormone [43]. In addition, a study analyzing the time to task failure during the different phases of the menstrual cycle on knee extensor muscles did not show any significant difference [44]. Additionally, previous authors have demonstrated that during a running protocol at 70% of VO_2_max, DOMS was increased during the luteal phase compared to the follicular phase, as witnessed by the increase in creatine kinase concentration. Our results therefore disagree with this finding, since DOMS did not demonstrate any significant difference throughout our protocol. This can be explained by the fact that the Interleukin-6 concentration in muscle, a protein that regulates muscle inflammation, is identical after a running protocol during the follicular and luteal phases [45]. Altogether, these findings could explain why RPE and sRPE were not impacted by the menstrual cycle in our study. Previous studies also demonstrated that perceptual responses (i.e., RPE) were not influenced by the different phases of the menstrual cycle [46]. In addition, training duration was kept identical throughout the protocol, explaining the absence of significant differences in sRPE. Therefore, combining performance indicators with psychological markers seems to suggest that the menstrual cycle is not the main cause of performance fluctuations.

In this paper, the findings could have been explained by the menstrual cycle determination method used. However, results demonstrating an impact of hormonal fluctuations on performance have been challenged by authors who criticized the determination methods of the different phases of the menstrual cycle [10]. These authors demonstrated that muscle contractile and aerobic capacities were not influenced by hormonal fluctuations using the clinical determination method [10]. In accordance with the majority of studies exploring and highlighting the impact of the menstrual cycle on performance, we chose to determine them based on the bleeding onset [22]. Therefore, we could have expected this method to significantly differentiate follicular and luteal phases, as in other studies using this determination method. Our results thus demonstrate that even using a simple menstrual cycle determination method, the impact of the menstrual cycle on physical and psychological performance is not clear.

This study has several limitations. The first limitation concerns the small sample size, which could explain the lack of significant effect concomitant with moderate effect sizes. A larger sample size, taking into account different training contexts and with different determination methods, would be of interest. For instance, additional professional teams combined with different training contents and periodization periods should be considered, taking into account individual responses. Moreover, the present experimental design should be replicated over a longer period with multiple menstrual cycles. In addition, the present experimental design studied the influence of the menstrual cycle on training without considering the competitive context of each week. Therefore, future studies should replicate this protocol, incorporating competitive workload as in other studies [16]. Furthermore, the menstrual cycle could be evaluated with hormonal measures to identify more precisely the different phases of the menstrual cycle and its interindividual variations.

## 5. Conclusions

Our results demonstrated that hormonal variations during the menstrual cycle do not influence the psychological and physical markers of performance. While performance could be expected to increase during the late follicular phase due to estrogen production, our results demonstrated comparable physical and psychological performance levels throughout the menstrual cycle. Therefore, the impact of the menstrual cycle on performance remains to be proven for academy elite women soccer players.

## Figures and Tables

**Figure 1 sports-12-00016-f001:**
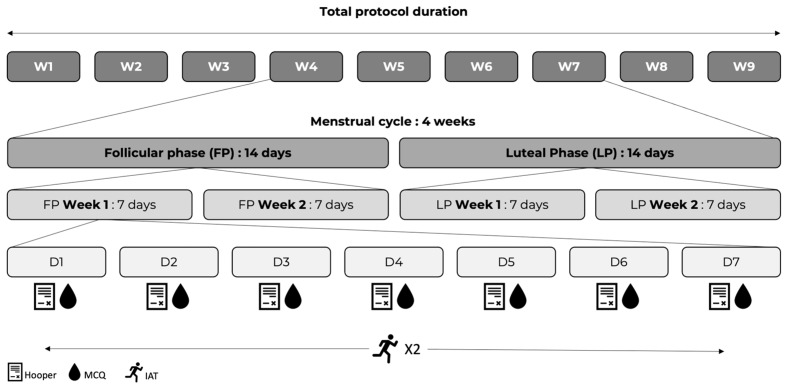
Experimental design. W: week; D: day; MCQ: menstrual cycle questionnaire; IAT: Illinois Agility Test. IAT was performed twice per week on varying days depending on athletes’ menstrual cycle onset.

**Table 1 sports-12-00016-t001:** Weekly organization and daily activities during the 9-week protocol.

		MD+1	MD+2	MD−4	MD−3	MD−2	MD−1	MD
Test	AM	Hooper + MCQ	Hooper + MCQ + IAT	Hooper + MCQ	Hooper + MCQ	Hooper + MCQ	Hooper + MCQ	Hooper + MCQ
	PM	-	-	-	-	IAT	-	-
Training	AM	-	Rondo (15 min)—Tactical work (20 min)—SSG (15 min) (60 min session)	-	-	-	-	-
	PM	-	Gym-based strength session (60 min session)	Plyometric work (20 min)—Positional work (30 min)—Tactical work (20 min)—SSG (25 min) (110 min session)	Aerobic intermittent work (20 min)—Finishing drills (15 min)—Tactical work (15 min)—LSG (30 min) (90 min session)	Dissociated speed work (20 min)—Finishing drills (25 min)—SSG (20 min) (75 min session)	-	Match

MD: match day; MCQ: menstrual cycle questionnaire; IAT: Illinois Agility Test; SSG: small-sided games; LSG: large-sided games.

**Table 2 sports-12-00016-t002:** Values of the monitored variables during weeks 1 and 2 of the follicular and luteal phases with *p*-values from the repeated measures ANOVA.

Indicator	FP Week 1	FP Week 2	LP Week 1	LP Week 2	Phase	Week	Phase × Week	$\eta_{p}^{2}$
Duration (min)	76.8 ± 2.9	75.3 ± 5.8	78.0 ± 4.9	71.1 ± 13.6	0.592	0.063	0.359	0.094
RPE (a.u.)	6.0 ± 0.3	6.1 ± 0.4	5.8 ± 0.6	6.0 ± 0.6	0.099	0.150	0.846	0.004
sRPE (a.u.)	462.5 ± 28.1	460.3 ± 54.8	449.9 ± 49.3	430.2 ± 72.2	0.313	0.205	0.645	0.025
IAT (s)	17.2 ± 0.5	17.3 ± 0.5	17.3 ± 0.6	17.3 ± 0.6	0.302	0.273	0.633	0.026
Fatigue (a.u.)	3.3 ± 0.6	3.2 ± 0.5	3.2 ± 0.7	2.9 ± 0.9	0.568	0.407	0.444	0.066
Stress (a.u.)	2.2 ± 0.9	2.0 ± 1.2	2.1 ± 1.3	1.8 ± 1.0	0.437	0.072	0.868	0.003
Sleep (a.u.)	2.3 ± 0.5	2.3 ± 0.7	2.3 ± 0.8	2.1 ± 0.9	0.570	0.373	0.398	0.080
DOMS (a.u.)	3.1 ± 0.8	2.8 ± 0.7	2.7 ± 0.9	2.3 ± 0.7	0.060	0.211	0.725	0.014
Hooper Index (a.u.)	10.7 ± 2.1	10.3 ± 2.1	10.3 ± 2.6	9.2 ± 2.4	0.088	0.150	0.403	0.079

Results for monitored variables are presented as mean values ± SD. FP: Follicular phase; LP: Luteal phase; RPE: Rate of perceived exertion; sRPE: Session rate of perceived exertion; IAT: Illinois Agility Test; DOMS: Delayed onset muscle soreness.

## Data Availability

The data presented in this study are available on request from the corresponding author.

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
