# Peer review of "Impact of the Menstrual Cycle on Physical Performance and Subjective Ratings in Elite Academy Women Soccer Players"

_sports, 2024, doi:10.3390/sports12010016_

Round 1

Reviewer 1 Report

Comments and Suggestions for Authors

---The study was well conducted, presents a fluid reading and presentation of results. However, some adjustments are necessary. I suggest that the authors meet all of them.

 Specific Comments 

-Introduction - It is not clear in the introduction which variable will be used or what the term "physical readiness" refers to. Only in the methods did the authors report that this parameter will be considered based on the Hooper questionnaire and Illinois Agility Test. Define this term better in the introduction, explaining the reasons why it is relevant to the modality, and, therefore, should be studied in line with the menstrual cycle.

-Line 74-75 – Define the criteria for the irregular cycle in the manuscript

-Line 114 – Provide a well elaborated figure with the experimental design.

-Line 146 – Provide the sample size estimation or calculate the statistical power.

-Line 146 – 153 – Was data presented as mean and standard deviation?

-Line 156 - Explain, in the manuscript, the large drop-out in the sample.

-Table 2 – Invert the order of appearance regarding the variables. In the first four columns present the mean and standard deviation, and then in the last four the p values and Np2

-The term “our results” appears in the begging of the first three paragraphs of the discussion. Improve this.

-Line 233 – “Indeed, the menstrual cycle is a complex biological process that players can respond to differently” This is not a limitation.

-Line 234 – 238 - This is vague and doesn't say much.

Author Response

-Introduction, it is not clear in the introduction which variable will be used or what the term "physical readiness" refers to. Only in the methods did the authors report that this parameter will be considered based on the Hooper questionnaire and Illinois Agility Test. Define this term better in the introduction, explaining the reasons why it is relevant to the modality, and, therefore, should be studied in line with the menstrual cycle.

Response. The authors thank the reviewer for the helpful comments that would help improving the quality of the present manuscript. All comments have been taken into account and all alterations have been highlighted in red font. Considering the first comment, authors, have first decided to limit the use of the term readiness level that was too large considering the present study. Second, authors have clarified in introduction (last paragraph) the different variables being tested.

-Line 74-75 – Define the criteria for the irregular cycle in the manuscript

Response. The authors thank the reviewer for his/her comment. As requested we defined the criteria for the irregular cycle in the manuscript. “During the last 6 months prior to the protocol, players who reported irregular cycles (absence, or more than one-week delay) were not included in the study, nor were those who used pills as contraceptive methods due to their impact on the menstrual cycle [19–21].”

-Line 114 – Provide a well-elaborated figure with the experimental design.

Response. As recommended by the reviewer, we provided a figure with the experimental design. Please see the revised version of the present study.

-Line 146 – Provide the sample size estimation or calculate the statistical power. 

Response. The authors thank the reviewer for his/her comment. Accordingly, we calculated the sample size estimation. “The sample size was calculated a priori using G*Power (version 3.1.9.6, free software available at https://www.psychologie.hhu.de/ arbeitsgruppen/allgemeine-psychologie-und-arbeitspsychologie/ gpower.html) using the following values: effect size of 0.3, power of 0.8, probability error of 0.05. A sample size of 17 volunteers was indicated.”

-Line 146 – 153 – Was data presented as mean and standard deviation?

Response. According to the reviewer’s comment, we added the following sentence: “Statistical analyses were conducted using JASP (version 0.14, JASP Team 2020, University of Amsterdam, available free at https://jasp-stats.org/download/ (accessed on 9 December 2022). Results are presented as mean values (± standard deviation, SD)."

-Line 156 - Explain, in the manuscript, the large drop-out in the sample. 

Response. The authors thank the reviewer for his/her comment. We explained the large drop-out: "The other 8 participants were lost due to at least one missed training session (injury (n=2), sickness (n=3)) or did not fill the MCQ in correctly (n=2)."

-Table 2 – Invert the order of appearance regarding the variables. In the first four columns present the mean and standard deviation, and then in the last four the p values and Np2

Response. According to the reviewer’s comment, we inverted the order of the columns in the table 2.

-The term “our results” appears in the begging of the first three paragraphs of the discussion. Improve this. 

Response. According to the reviewer’s comment we improved the beginning of our paragraphs in the discussion section.

-Line 233 – “Indeed, the menstrual cycle is a complex biological process that players can respond to differently” This is not a limitation. 

-Line 234 – 238 - This is vague and doesn't say much.

Response for these two comments. As recommended by the reviewer, we deleted the sentence in the limitation section and the limitation section has slightly been modified. "This study has several limitations. The first limitation concerns the small sample size which could explain the lack of significant effect concomitant with moderate effect sizes. A larger sample size taking into account different training contexts, and with different determination methods would be of interest. For instance, additional professional teams combined with different training contents and periodization periods should be considered, taking into account the individual responses. Moreover, the present experimental design should be replicated over a longer period with multiple menstrual cycles."

Reviewer 2 Report

Comments and Suggestions for Authors

I appreciate your efforts. Working with teams is hard work to get responses. I hope my comments help.

Line 15 – perhaps Questionnaire needs to be capitalized as you wrote Hooper Questionnaire in the methods section.

Line 27 – Nowadays seems a bit informal. It is just a thought to start with The influence of…

Line 56 – I found this review and a few other manuscripts with a quick search in Google Scholar with the terms - menstrual cycle and elite soccer. I understand your point is elite soccer. But it seems there is more research out there. I believe you should incorporate more of it.

https://www.frontiersin.org/articles/10.3389/fphys.2021.654585/full

Ross Julian, Sabrina Skorski, Anne Hecksteden, Christina Pfeifer, Paul S Bradley, Emiel Schulze & Tim Meyer (2021) Menstrual cycle phase and elite female soccer match-play: influence on various physical performance outputs, Science and Medicine in Football, 5:2, 97-104, DOI: 10.1080/24733938.2020.1802057

The Influence of Menstrual Cycle on Bioimpedance Vector Patterns, Performance, and Flexibility in Elite Soccer Players 2021. International Journal of Sports Physiology and Performance. DOI: https://doi.org/10.1123/ijspp.2021-0135

Results – Table 2

There are not significant p values, but some of the partial eta-squared values are at least medium in meaningfulness. Are there meaningful results being that being masked by the p values and small sample? This seems important to double check. I did not read anything about meaningfulness in your discussion.

Line 192-220 – this paragraph seems extra-long.

Line 233 – sample size seems to be the first limitation to be discussed.

More elite soccer publications need to be incorporated into your results and even with sub elite. In short, there is literature out there it seems.

Comments on the Quality of English Language

Minor edits needed.

Author Response

I appreciate your efforts. Working with teams is hard work to get responses. I hope my comments help.

Response. The authors thank the reviewer for the helpful comments that would help improving the quality of the present manuscript. All comments have been taken into account and all alterations have been highlighted in red font.

Line 15 – perhaps Questionnaire needs to be capitalized as you wrote Hooper Questionnaire in the methods section.

Response. We thank the reviewer for his comment. The word “questionnaire” has been capitalized. “Players answered the Hooper Questionnaire daily”

Line 27 – Nowadays seems a bit informal. It is just a thought to start with The influence of…

Response. As recommended by the reviewer, we removed the word “nowadays”: “The influence of the menstrual cycle”

Line 56 – I found this review and a few other manuscripts with a quick search in Google Scholar with the terms - menstrual cycle and elite soccer. I understand your point is elite soccer. But it seems there is more research out there. I believe you should incorporate more of it.

https://www.frontiersin.org/articles/10.3389/fphys.2021.654585/full

Ross Julian, Sabrina Skorski, Anne Hecksteden, Christina Pfeifer, Paul S Bradley, Emiel Schulze& Tim Meyer (2021) Menstrual cycle phase and elite female soccer match-play: influence on various physical performance outputs, Science and Medicine in Football, 5:2, 97-104, DOI: 10.1080/24733938.2020.1802057

The Influence of Menstrual Cycle on Bioimpedance Vector Patterns, Performance, and Flexibility in Elite Soccer Players 2021. International Journal of Sports Physiology and Performance. DOI: https://doi.org/10.1123/ijspp.2021-0135 

Response. According to the reviewer’s comments, we incorporated the proposed manuscripts within our paper in the introduction and the discussion section. "The impact of the menstrual cycle on performance in such a population should, therefore be explored. However, the scientific literature is still scarce concerning its impact on elite soccer players [16], or inconclusive [17–19]." … "Yet, using field tests as in our protocol, authors demonstrated the absence of any impact of the menstrual cycle on countermovement jump, 20m, and 30m sprint performances [18,36]."

Results – Table 2

There are not significant p values, but some of the partial eta-squared values are at least medium in meaningfulness. Are there meaningful results being that being masked by the p values and small sample? This seems important to double check. I did not read anything about meaningfulness in your discussion.

Response. Effect sizes have been double-checked. No error has been noticed. We confirm the lack of any significant ANOVA effect that could be associated with moderate effect sizes for training duration, fatigue, sleep and global Hooper index. These effect sizes are for the general ANOVA and are not related to a pairwise comparison between phases or weeks. Thus, and because no significant effect was obtained, authors cannot discuss these values without being highly speculative. The only possibility is to acknowledge for additional volunteers in the limitation section.

Line 192-220 – this paragraph seems extra-long.

Response. We thank the reviewer for his/her comment. While we understand that the paragraph might seem long, we think that reducing it would diminish the quality of the information and therefore decrease the quality of the paper.

Line 233 – sample size seems to be the first limitation to be discussed.

Response. As recommended by the reviewer, we changed the limitation section. “This study has several limitations. The first limitation concerns the small sample size which could explain the lack of significant effect concomitant with moderate effect sizes. A larger sample size taking into account different training contexts, and with different determination methods would be of interest. For instance, additional professional teams combined with different training contents and periodization periods should be considered, taking into account the individual responses. Moreover, the present experimental design should be replicated over a longer period with multiple menstrual cycles."

More elite soccer publications need to be incorporated into your results and even with sub elite. In short, there is literature out there it seems.

Response. In accordance with the reviewer’s request, we added references concerning the menstrual cycle.

Round 2

Reviewer 1 Report

Comments and Suggestions for Authors

I have no further comments. 

Author Response

Thank you for all interesting comments

Reviewer 2 Report

Comments and Suggestions for Authors

Dear authors,

Thank you for your revised manuscript. I appreciate you efforts. I would like for you to indicate in your abstract that of the 18 females, 10 completed all requirements.

I hope you continue your important work in female athletics.

Comments on the Quality of English Language

Just fine.

Author Response

Authors thank the reviewer for helping improving the present manuscript. According to reviewer comment, authors have added the requested information in abstract.